# Resorbable Biomaterials Used for 3D Scaffolds in Tissue Engineering: A Review

**DOI:** 10.3390/ma16124267

**Published:** 2023-06-08

**Authors:** Sara Vach Agocsova, Martina Culenova, Ivana Birova, Leona Omanikova, Barbora Moncmanova, Lubos Danisovic, Stanislav Ziaran, Dusan Bakos, Pavol Alexy

**Affiliations:** 1Institute of Natural and Synthetic Polymers, Faculty of Chemical and Food Technology, Slovak University of Technology, 812 37 Bratislava, Slovakia; sara.agocsova@stuba.sk (S.V.A.); barbora.moncmanova@stuba.sk (B.M.); dusan.bakos@stuba.sk (D.B.); pavol.alexy@stuba.sk (P.A.); 2Panara a.s., Krskanska 21, 949 05 Nitra, Slovakia; ivana.birova@panara.eu (I.B.); leona.omanikova@stuba.sk (L.O.); 3National Institute of Rheumatic Diseases, Nabrezie I. Krasku 4, 921 12 Piestany, Slovakia; lubos.danisovic@fmed.uniba.sk (L.D.); stanoziaran@gmail.com (S.Z.); 4Institute of Medical Biology, Genetics and Clinical Genetics, Faculty of Medicine, Comenius University in Bratislava, 811 08 Bratislava, Slovakia; 5Department of Urology, Faculty of Medicine, Comenius University, Limbova 5, 833 05 Bratislava, Slovakia

**Keywords:** biomaterials, biopolymers, bioceramics, hybrid biomaterials, 3D scaffolds, tissue engineering, foreign body response, immunoengineering

## Abstract

This article provides a thorough overview of the available resorbable biomaterials appropriate for producing replacements for damaged tissues. In addition, their various properties and application possibilities are discussed as well. Biomaterials are fundamental components in tissue engineering (TE) of scaffolds and play a critical role. They need to exhibit biocompatibility, bioactivity, biodegradability, and non-toxicity, to ensure their ability to function effectively with an appropriate host response. With ongoing research and advancements in biomaterials for medical implants, the objective of this review is to explore recently developed implantable scaffold materials for various tissues. The categorization of biomaterials in this paper includes fossil-based materials (e.g., PCL, PVA, PU, PEG, and PPF), natural or bio-based materials (e.g., HA, PLA, PHB, PHBV, chitosan, fibrin, collagen, starch, and hydrogels), and hybrid biomaterials (e.g., PCL/PLA, PCL/PEG, PLA/PEG, PLA/PHB PCL/collagen, PCL/chitosan, PCL/starch, and PLA/bioceramics). The application of these biomaterials in both hard and soft TE is considered, with a particular focus on their physicochemical, mechanical, and biological properties. Furthermore, the interactions between scaffolds and the host immune system in the context of scaffold-driven tissue regeneration are discussed. Additionally, the article briefly mentions the concept of in situ TE, which leverages the self-renewal capacities of affected tissues and highlights the crucial role played by biopolymer-based scaffolds in this strategy.

## 1. Introduction

Increasing demand for organs and tissues, driven by the need to address limitations associated with autografts and allografts, has stimulated the development of TE as a field of scientific inquiry. In recent years, the discipline of materials science in TE has amalgamated an engineering-oriented approach with knowledge derived from the natural sciences and medicine. The objective has been to fabricate biological substitutes capable of restoring, securing, or enhancing the functionality of damaged tissues or organs [1]. Biomaterials represent the basic components and play a crucial role in scaffold engineering [2]. Biocompatibility is an essential prerequisite for biomaterials, encompassing specific mechanical, biochemical, and physical properties [3]. Biocompatibility, as defined broadly, denotes the ability of a material to interact with living cells or tissues without inducing toxicity or provoking immunological responses. Given that biomaterial cell interactions significantly influence cell viability, primarily through mechanisms such as proliferation and differentiation, the characteristics of biomaterials, including surface chemistry, charge, roughness, reactivity, hydrophilicity, and rigidity, necessitate careful consideration [4]. A scaffold, which constitutes a three-dimensional (3D) mesh composed of diverse biocompatible, bioactive, and biodegradable materials, assumes the role of providing structural support to the attached cells and creating an appropriate environment conducive to their proliferation, migration, and differentiation [5]. The primary objective of scaffolds is to mimic the structure and function of the extracellular matrix (ECM) of the original tissue [6].

An ideal scaffold must possess suitable physical, chemical, and biological properties to facilitate cell proliferation, migration, and tissue formation within 3D space. These structures should enable adequate oxygenation, nutrient supply, and removal of metabolic waste during tissue regeneration. Furthermore, scaffolds should exhibit the ability to withstand mechanical stress [7,8]. The presence of a porous structure favors effective cell colonization (Figure 1a,b). Additionally, the surface structure and chemistry of these polymeric matrices need to be taken into account to ensure optimal cell attachment. 

Scaffolds can be engineered with bio-based and fossil-based polymers, or with bio-based ceramics with suitable combinations [9]. Bio-based polymer scaffolds generally exhibit minimal immune responses and promote favorable cell interactions, while fossil-based polymers are comparatively more economical, possess greater strength and offer improved functionality. However, they may exert toxic effects and elicit adverse immune responses. Polymers, in general, serve as widely utilized biomaterials for constructing soft matrices used in the production of highly transplanted organs such as the kidneys and liver. Furthermore, they have successfully found applications in the regeneration of muscles, tendons, heart valves, arteries, bladder, and pancreas [10]. Hybrid biomaterials, comprising combinations of two or more biomaterials, exhibit enhanced functionalities and can address the limitations associated with individual materials, thereby satisfying a broad range of clinical requirements. The subsequent text provides a comprehensive analysis of recent literature focused on biopolymers within the context of TE. Key parameters necessary for scaffold design are also discussed. Notably, the interaction between biomaterial-based scaffolds and the host organism is described, as this phenomenon is deemed crucial for their successful translation into clinical medicine.

## 2. Materials Used for 3D Scaffold Engineering

### 2.1. Fossil-Based Polymers

Fossil-based polymers (FBPs) are polymers that are artificially produced under laboratory or industrial conditions from fossil raw materials, usually petroleum or natural gas. They are man-made and assembled by chemical reactions. Representatives that can be used in TE are poly-(ε-caprolactone) (PCL), poly(vinyl) alcohol (PVA), polyurethane (PU), poly-(propylene fumarate) (PPF), and polyethylene glycol (PEG) (Table 1). They usually have a controlled structure, no immunological reaction, and a higher degree of processing flexibility. They can also provide better control over the synthesis and design of the polymer itself [11]. FBPs have several advantages for regenerative medicine applications. They can be produced in reproducible high quality and purity. In addition, they have good mechanical properties and thermal resistance compared to natural polymers [12]. FBPs are widely used in TE, and their chemical, physical, mechanical, and morphological properties (e.g., pore size and distribution, surface texture, typology, and crosslinking) can be adjusted according to the intended application (Figure 2) [13]. The chemical structure of FBPs determines their degradability and other properties that are important for scaffold design. These polymers can form unlimited structures and shapes, which is a great advantage. The degradation of polymers used in TE is mainly by hydrolysis. The rate of degradation can be modulated by adjusting the composition, molecular weight, and end groups in the polymer, as well as by the composition and geometry of the device [14]. Moreover, they can be easily engineered and chemically modified. These properties present important requirements in the context of TE and subsequent medical applications. Compared to natural biodegradable polymers, bioresorbable FBPs offer better controlled mechanical properties while biodegradability is maintained. One of their major advantages is their customizable porosity, large-scale production, and long shelf life [15]. These polymers can be classified according to their hydrophilicity, hydrophobicity, and degradability [16].

#### 2.1.1. Poly(ε-caprolactone) (PCL)

PCL is an extremely tough aliphatic polyester and is widely used in various biomedical applications [16]. It is a polymer with repeating hexanoate units and has a low melting point of 60 °C. PCL is bioresorbable and has good stiffness, mechanical elasticity, thermal stability, biocompatibility, and rheological and viscoelastic properties. It has low decomposition rates and is highly flammable, non-toxic, and inexpensive [16]. On the other hand, poor cell adhesion and slow degradability are disadvantages of this synthetic biopolymer [17]. To minimize these limitations, the combination of PLC with other polymers, e.g., poly-lactic acid (PLA), has been developed to create a less hydrophobic construct with better mechanical properties and improved degradability. When applied in vivo, PCL proves to be stable and shows no visible degradation after 6 months [18]. PCL has been used in the production of hard (bone and dental applications) and soft tissues (skin, liver, cartilage, ligaments, muscle, nerves, retina, and blood vessels) [17]. In the context of skin TE, studies have demonstrated that a layer-by-layer method is suitable for modeling PCL scaffolds. These provide a suitable matrix for the fabrication of cell-colonized constructs consisting of epidermal and dermal skin layers. Experiments also showed that the proliferation and survival rates of fibroblasts were sufficient, allowing skin regeneration [16]. Porous PCL scaffolds can also be utilized in vivo. Their long-term degradation behavior is very favorable for their in vivo performance, especially in the context of hard tissue TE. Studies have described that the optimal porosity for PCL scaffolds should be greater than 90% to effectively control the rate of degradation and provide an appropriate environment for cell attachment, proliferation, and subsequent differentiation [12].

#### 2.1.2. Poly(vinyl) Alcohol (PVA)

PVA can be utilized in various biomedical applications. Due to its properties, such as good biodegradability, biocompatibility, hydrophilicity, permeability, flexibility, and the possibility to be blended with other biopolymers, PVA is an attractive polymer for scaffold engineering. It is often blended with chitosan, or polyhydroxy butyrate to produce nanofibers for wound healing [16]. In addition, studies have described the possibility of re-epithelization of PVA nanofibers prepared by electrospinning, which were subsequently mixed with Carica papaya and gelatin. This phenomenon happened due to the moist environment provided by this composite scaffold [20]. PVA nanofibers also show significant anti-bacterial activity against *Staphylococcus aureus* (gram-positive) and *Escherichia coli* (gram-negative), which is important for the healing process and confirms their potential for wound healing applications [21]. PVA is also typical for its ability to store a significant amount of water or biological fluids and to act as a lubricant resembling the cartilage surface [22]. Another reason why it attracts attention in the field of artificial cartilage TE is its elastic physical properties and low coefficient of friction [23,24]. Disadvantages of PVA-based scaffolds are biodegradability, which is lower after crosslinking, and reduced cell adhesion due to the hydrophilic moieties (i.e., hydroxyl groups) presented on the polymer scaffold. Combining PVA with other synthetic or natural materials could be the solution for these drawbacks [25]. PVA is soluble in water and therefore needs to be crosslinked to reduce its solubility and improve mechanical properties. PVA-based scaffold design with >85% is favorable for the TE applications [14]. 

#### 2.1.3. Polyethylene Glycol (PEG)

PEG (polymer of ethylene oxide) is known as polyethylene oxide (PEO) or polyoxyethylene [16]. PEG is a water-soluble, non-ionic, and biodegradable synthetic polymer that has a wide range of applications in TE [29]. The mechanical properties of composites can be improved by the good mechanical stiffness of PEG [30]. PEG exhibits low immunogenicity and nontoxicity [29]. Moreover, PEG-based scaffolds present hydrated structures that improve cytocompatibility [31]. As mentioned above, mechanical stiffness is one of the most important properties. This phenomenon can be improved by chemical crosslinking [32].

#### 2.1.4. Polypropylene Fumarate (PPF) 

PPF is a linear polyester composed of fumaric acid and propylene glycol as monomer units. This biodegradable polyester offers excellent strength due to the crosslinking of double bonds to form a polymeric network [33]. This characteristic is ideal for its orthopedic applications. It can be combined with PEG, PLA, or PCL to improve their hydrophobicity [34]. Degradation of PPF occurs by hydrolysis of the ester bonds, and the rate of degradation depends on the molecular mass of the main chain, used crosslinkers, and the crosslink density [35]. PPF is mainly used in orthopedic applications because of its extremely good mechanical strength, good biocompatibility, and osteoconductivity. The aforementioned properties pre-determine PPF-based scaffolds for use in bone TE. Moreover, these matrices have also been described to degrade in the time frame relevant to bone healing and remodeling [36]. Moreover, PPF-based scaffolds have been investigated for their potential use in ophthalmology, cardiac, and neural TE applications [35].

#### 2.1.5. Polyurethane (PU) and Modified Polyurethanes (MPUs)

PU holds attractive properties as a framework material for TE applications. This is due to its wide range of versatile mechanical, physical, and biological characteristics, along with good biodegradability, biocompatibility, and high flexural strength [40]. Typical advantageous qualities include elasticity and the ability to convert to poly(ester-urethane) urea [16]. In addition, PU exhibits low cytotoxicity, interfacial tension, high thrombo-resistance, oxygen permeability, and suitable mechanical properties for various pharmaceutical and biomedical applications, such as wound dressing materials, drug carriers, and antimicrobial filters [40]. The bioactive behavior of PU scaffolds can be enhanced by biodegradable, electroactive, surface–functionalized products, ceramic, and natural polymers, leading to the formation of modified polyurethanes (MPUs) [41]. PU and MPUs can be used for both soft and hard TE [40]. 

FBPs have numerous applications in TE and represent one of the main groups within the polymers used for 3D scaffolds. However, current trends in scaffold fabrication are more focused on natural or bio-based polymers. These are derived from renewable resources such as plant waste and algae and are, therefore, more environmentally friendly. In addition, their biological background is more favorable in the context of in vivo application. Therefore, the majority of the text in this article is oriented on natural and bio-based polymers. 

### 2.2. Natural and Bio-Based Polymers

Natural polymers (biopolymers) are obtained from natural materials and provide a guarantee of natural structure, biomimetic and bioactive nature [42]. Bio-based polymers can be constructed in a synthetic way based on monomers originating in biomass, obtained by extraction or the transformation of biomass sources into monomers for polymerization of new polymers, which naturally do not exist in nature. PLA is a typical example of such a bio-based polymer. Biopolymers often contain multiple polysaccharides (chitosan, alginate, and cellulose) and polypeptides (gelatin, fibrin, and silk). In addition, many functional groups have recently been introduced into biopolymers to modify their physical and physiological properties, making them multifunctional and smart [43]. They fulfill diverse functions in their natural environment. Biocompatible and bioresorbable natural polymers that mimic the ECM are preferred over synthetic ones in the context of scaffolding in TE. Their degradation should be in balance with the formation of new tissue. Table 2 offers the summarization of their basic properties together with the most common TE applications.

#### 2.2.1. Collagen

With no immunological or inflammatory response, collagen is a biocompatible biopolymer that is simple to produce. It also has a porous structure with interconnecting pores. It is also crucial for preserving the biological and structural integrity of ECM. The development and wound healing of organs and tissues, including skin, bone, the neurological system, and the vascular system, have all benefited from its exceptional biocompatibility and bioresorbability [50,51]. 

Collagen fibrils are the basic supporting element in the connective tissues. There are seven collagens that can produce fibrils (types I–V, types XI, XXIV, and XXVII). Type I is the quantitatively predominant fibrillar collagen in vertebrates [52]. The D-periodic spacing and orientation of the modified collagen fibers (MCF) serve as typical representations of the morphology of the collagen network. Open sites for mineral nucleation, proteoglycan binding, and crosslinking are provided by the D-periodic spacing of MCF. The deposition of minerals is facilitated by the appropriate arrangement of collagen fibrils, which also enhances the bone’s mechanical qualities [53]. Because of the mentioned properties, this polymer found its applications in TE of bone grafts (Figure 3) [54]. 

Standard use of collagen graft substitutes involves seeding them with mesenchymal stem cells, which have the capacity for osteogenic differentiation [55]. Additionally, numerous investigations on collagen scaffolds for endometrial regeneration have been conducted. Li et al. created collagen scaffolds that were targeted with human basic fibroblast growth factor (bFGF), for instance. After 90 days, endometrial cells completely replaced the collagen/bFGF scaffold. This phenomenon was determined by the results of hematoxylin and eosin staining [56].

#### 2.2.2. Hyaluronic Acid (HA)

HA is a linear polyanionic polysaccharide, with the structure of glucuronic acid and N-acetylglucosamine repeating units linked via alternating β-1,4 and β-1,3 glycosidic bonds, which occurs naturally in all living organisms and maintains an appropriate level of tissue hydration [57]. It is one of the main components of the ECM and can be used to crosslink collagen fibrils, as it participates in mineralization [58]. In addition, HA is biocompatible, bioresorbable, and flexible in composition and structure. It has been widely used to produce hydrogel composite materials for bone TE and oral disease treatment [59].

HA affects various physiological processes, which are dependent on its molecular weight. High molecular weight HA inhibits cell proliferation and migration, while low molecular weight HA is involved in cell reproduction [60]. In the context of TE applications, HA is commonly modified with various chemical groups (e.g., methacrylate) to improve its mechanical properties as a hydrogel network [61]. Studies showed that hydrogels with a middle pore size of 200–250 μm supported the best cell proliferation and furthest 3D migration [62]. Another study determined that composite scaffolds based on collagen and HA enhanced the biological properties of seeded cells, which resulted in a higher proliferation and differentiation rate, leading to bone regeneration [63].

#### 2.2.3. Chitosan, Fibrin

In addition to collagen and HA, two other natural polymers, chitosan and fibrin, are being investigated for their use in TE. These polymers form fibers or foams. Fibrin naturally assembles into gels and thrombus formation in vivo during wound healing and is an important component of the ECM in the body [64]. Chitosan is derived from chitin and is a suitable and promising biomaterial for articular cartilage regeneration. However, chitosan is poorly soluble in water under physiological conditions, which limits its widespread use in TE [44]. The hydroxyl and amino groups of the polymer provide several options for derivatization or grafting of the desired bioactive groups, and the pH-dependent solubility of chitosan allows the use of relatively mild processing methods. This property is particularly important if the incorporation of bioactive substances is desired before the formation of the 3D microstructure [45].

In the fibrin scaffold, pores can be distributed in the range of 0–35.86 µm and 35.86–89.65 µm, with a total porosity of 43.28% [65]. Chitosan has good biocompatibility and bioresorbability, making it a suitable material for TE applications. There is not much-related research on the direct application of chitosan in 3D scaffolds yet. However, experiments mainly concern 3D composite scaffolds that are mixed with other biomaterials for permanent implantation in the human body [46]. The chitosan–alginate (CA) complex scaffold has numerous benefits over 3D synthetic polymers and protein matrices because of its natural origin. When compared to their equivalents, CA scaffolds have a noticeably higher mechanical strength due to the ionic interaction of the amine group of chitosan with the carboxyl group of alginates [47]. Alginate and chitosan have reduced immunogenicity and demonstrated structural stability as well [48]. Compared to other natural polymers, CA scaffolds can be generated from solutions with a physiological pH, making it possible to synthesize them while uniformly incorporating growth inputs with a reduced risk of denaturation. For both in vitro and in vivo stem cell expansion and differentiation, growth factors are encapsulated in the scaffold matrix. The rate of scaffold bioresorbability, which is controlled by scaffold synthesis conditions, determines the degree of growth factor release. This approach can potentially enable the direct implantation of stem cell-populated scaffolds for a variety of applications in TE and regenerative medicine, in addition to providing a clean environment for stem cell regeneration [49].

#### 2.2.4. Polysaccharides

Polysaccharides have good bioresorbability, biocompatibility, easy derivatization, and low production costs. They offer similarities to the ECM, making them a suitable candidate for TE. Structured polysaccharides and stored polysaccharides are two different categories of polysaccharides. Chitin, found in crab shells, is an example of a structural polysaccharide, whereas glycogen and starch are examples of storage polysaccharides. Natural polymers have several benefits, but they also have drawbacks related to their branching, sequencing, and molecular weight distribution that make it difficult to create uniform scaffolds without negatively impacting biorecognition and rheology [107].

Gram-negative bacteria such as Xanthomonas bacteria can produce xanthan gum (XG), which is an anionic polysaccharide. Its side chains are made up of D-mannose and D-glucuronic acid in a 2:1 ratio, and its backbone is the same as that of cellulose [66]. Its molecular mass is between 2106 and 2107 Da. Because of hydrogen bonds, the main chain’s helical shape coils around the side chain [67]. Since XG, in its purest form, is bioresorbable and biocompatible, it is frequently employed in drug administration, bone regeneration, wound healing, and soft TE. Therefore, there is a demand for biomaterials with possibly better qualities. Recently, nanocomposite fibers using chitosan and XG were created to transmit bioactive compounds [68]. After being crosslinked, XG, a natural substance with good biocompatibility, can become a soft hydrogel. In the realm of spinal cord repair, freeze-drying technology has made it possible to produce porous structures quickly and easily [69]. In this instance, the gel scaffold was created using a freeze-drying process after XG and graphene oxide were joined through metal coordination and hydrogen bonding. This method creates a scaffold that mimics the spinal cord characteristics and can preserve the milieu for cell growth, while also resisting compression of the surrounding tissue. For the regeneration of spinal cord tissue, the porous interior 3D structure resembles the ECM and offers growing room and nutrient transport channels. Studies conducted under both laboratory and in vivo conditions (animal model) showed that the electroconductive, porous, and soft gel scaffold had high biocompatibility, could reduce the growth of astrocytes around the site of spinal cord damage, and helped wounded rats to regain their ability to move. 

Among the various polysaccharides, dextran has also been successfully used in TE applications. It is a hydrophilic carbohydrate biopolymer that decomposes inappropriate physical environments without any effect on cell viability [70]. Dextran consists of branched polysaccharides of repeating α-linked d-glucopyranosyl units of various lengths and degrees of branching. There are various methods to chemically crosslink dextran to form a hydrogel [71]. Dextran hydrogels can provide excellent conditions for cell proliferation. However, their comparatively low biological activity limits their use in hard TE applications. The bioactivity of dextran hydrogel networks can be improved by incorporating bioceramics into the matrix [72].

Starch belongs to the group of natural polymers and is widely used in biomedical applications due to its significant bioresorbability, biocompatibility, availability, renewability, and easy processing (Figure 4). This carbohydrate consists of many glucose units connected by glycosidic bonds [73]. The 6–8 μm voids between starch agglomerates account for about 20% of the total porosity; the 1–35 μm voids around starch granules account for about 60%; and the very small pores account for 10%. Possible causes of the remaining 10% of the overall porosity are pores larger than 150 μm [74]. Starch is capable of oxidation and reduction and can precipitate when forming hydrogen bonds. Its granules exhibit hydrophilic properties and chain intermolecular association due to the hydrogen bonds formed by hydroxyl groups on the surface of the granules. The hydrophilicity of starch can be used to improve the degradation rate of some degradable hydrophobic polymers. This natural polymer is completely degradable and bioresorbable in a wide variety of environments and can be hydrolyzed to glucose by microorganisms or enzymes. On the other hand, it is not suitable for direct use because of its low dimensional stability and mechanical properties [75]. When combined with hydroxyapatite in the context of bone tissue repair, the scaffold’s improved structural and biological functions were described [76]. In addition, the bioresorbable properties of the bone scaffold play an important role in the regeneration of affected bone tissue. Biodegradable matrices recommended for hard tissue TE must be uniformly degraded in the human body during bone mineralization to improve bone tissue replacement and growth processes [77]. The advantage of using a natural polymer (starch) as an organic component in a bone scaffold is its ability to adapt its bioresorbability properties. The hydroxyl groups in the starch make the scaffold’s surface more hydrophilic and thus improve the hydrophilicity of the hydroxyapatite. Starch hydroxyl groups also increase the attraction of ions (phosphate and calcium) of the bone skeleton, which can speed up and make the apatite nucleation and subsequent bone mineralization faster and more efficient, which improves the overall regeneration of the bone tissue [78].

#### 2.2.5. Poly (Lactic Acid) (PLA)

PLA is a thermoplastic polymer characterized by high mechanical resistance, suitable biocompatibility, and bioresorbability. It is made from renewable and non-toxic sources of raw materials [79]. Lactic acid (LA) is converted into PLA through condensation polymerization or ring opening. Due to the chiral nature of LA and the two asymmetric centers that it has, it can be formed in a wide variety of forms and also have the following isomers: L, D, and D, L isomers, as well as the D isomer [80]. Amorphous polymers are produced by a higher share of D monomers (>15%) and more crystalline PLA by using high L monomer concentrations. While D, L-lactide produces the amorphous structure of poly (D, L-lactide), highly pure L- and D-lactides create semicrystalline polymers, such as poly (L-lactide) (PLLA) and poly (D-lactide) (PDLA). With varying distributions of the isomers in the structure, PLA can be produced in a variety of molecular weights between several thousand and several million [81].

PLA has been used in different biomedical applications, as it is fully FDA-approved [82]. PLA is suitable as a material for various medical uses and applications, such as surgical sutures, orthopedic and cardiovascular implants, drug delivery systems, and more [83]. In the context of TE, PLA satisfies the requirements to produce scaffold-based implants, since it is a bioabsorbable polymer that is hydrolytically degradable and does not form hazardous compounds while being broken down (Figure 5) [84]. Due to their outstanding biocompatibility, bioresorbable polymers such as PLA, PGA, and polycaprolactone (PCL), as well as their copolymers, are now often employed in biomedical devices [85]. These polymers are broken down without the use of any enzymes by straightforward hydrolysis of the ester bonds in their chains [86]. Following hydrolysis, the breakdown process’s byproducts are changed into non-toxic byproducts that can be reduced and eliminated by regular cellular function and urine [87]. The ideal pore diameter for the PLA scaffold is thought to be between 100 and 300 μm, while other studies have found that the presence of micropores (10 to 60 μm) promotes better cell proliferation. The choice of pore size depends on the method used to construct the scaffold and may have an impact on its mechanical and dynamic stability [88].

Since roughly 30 years ago, bioresorbable LA polyesters have been employed in surgery as bone fixation and suture materials. For the sustained delivery of bioactive compounds, LA and glycolic acid were already proposed as bioresorbable matrices in 1973 [89]. It has been demonstrated that PLA is a practical contender for resorbable plates and screws. As soluble suture meshes, bioabsorbable fixation devices are frequently utilized. As the implant bio-resorbs, patients do not need a second surgery to remove it, which lowers healthcare expenses and enables a progressive restoration of tissue function [84]. However, PLA’s hydrophobic nature and the absence of functional groups make it difficult for cells to adhere to this polymer. Another disadvantage is its slow hydrolytic degradation. However, the mechanical properties can be improved using various methods, such as blending with other bioresorbable polymers, forming composites, and copolymerization [90]. PLA blending with other bio-based materials or FBP with better wettability and faster bioresorbability provides an effective solution to modulate its biodegradability, with respect to the time required for tissue growth and regeneration [91].

#### 2.2.6. Polyhydroxyalkanoates (PHAs)

PHAs are linear polyesters that are hydrophobic and bioresorbable and are mostly made by microbes. In general, polyhydroxyalkanoate is a chemical compound made from a hydroxyl-alkanoic acid, also known as a carboxylic acid, which is an acid with the chemical formula HO-R-COOH [108]. Depending on the type of microorganism, growing circumstances, and method of polymer extraction, they can differ in molecular weight, structure, and content [109,110]. The morphology and crystal structure of these materials can be modified and influenced by mixing with additives, mainly nucleating agents, suitable plasticizers, and fillers, and are expected to significantly affect physical properties such as thermal, mechanical characteristics, and bioresorbability [111]. Naturally occurring PHAs are ubiquitous components of animal cell membranes. Low-molecular-weight poly-3-hydroxybutyrate (P3HB) is present in relatively high concentrations in human blood, and 3-hydroxybutyric acid (3HB) is a naturally occurring human metabolite found in the heart, liver, brain, lungs, and muscle tissue [92]. PHAs are used alone or in combination with other biological polymers for biomedical applications such as sutures, tendon repair devices, cardiovascular patches, TE scaffolds, orthopedic pins, adhesion barriers, and adhesion patches. PHAs also have excellent inherent biocompatibility, similar to other biological polymers [112]. Additionally, numerous other factors of chemical synthesis, including bulk material processing, surface treatment, sterilization, and adding a plasticizer to the polymer matrix, have a significant impact on the biocompatibility of the final medical implant. Therefore, PHAs are interesting subjects to be intensively evaluated under in vitro and in vivo conditions [113].

Polyhyroxybutyrate (PHB) is a member of the PHA group and shows a good degree of biocompatibility with various cells. In addition, it can be produced by many types of microorganisms due to its bacterial origin [92]. As a bioresorbable low-molecular-weight aliphatic polyester, PHB belongs to the most widely used biocompatible PHAs with an isotactic semicrystalline structure [93]. The degradation product of PHB is 3HB, which is non-toxic and, as mentioned above, a natural metabolite occurring in multiple tissues in the human body [92].

The nanoparticle form of PHB can affect the properties of the engineered scaffolds. In addition, crystallinity also significantly influences the biological behavior of the seeded cells, especially cell proliferation [94]. The pore size of PHB-based scaffolds is another crucial parameter that highly affects cell attachment, proliferation, and differentiation [95]. Therefore, new methods and techniques are still being developed to synthesize porous materials with optimal pore sizes. The larger surface area and good pore volume of the scaffold provide increased sites for initial cell insertion and attachment, resulting in enhanced cell adhesion to the scaffold surface [94]. Although PHB is stiff and brittle, with an elongation at break typically less than 5%, its thermoplastic polymer can easily be processed into textiles with good flexibility. PHB textiles have been extensively studied for applications in soft TE [96]. Tubular PHB textiles are also used to promote tissue healing in large vessels and hollow organs [97]. Due to its mechanical strength, PHB is also used in the production of bone plates and fillers [92]. In addition, recent studies have revealed the inherent piezoelectric properties of PHB, which show great potential to modulate cellular activity and thereby enhance bone tissue regeneration [98]. However, some disadvantages of PHB, such as low hydrophilicity, high crystallinity, stiffness, low bioresorbability rate, and high brittleness, suppress its application [99]. On the other hand, its blending with natural polymers such as keratin, gelatin, chitosan, starch, or lignin can improve its physical and biological properties [100].

Poly(3-hydroxybutyrate-co-3-hydroxyvalerate) (PHBV) is a thermoplastic, aliphatic polyester that also belongs to the PHAs group. This polymer has been known for many years, but especially in the last decade, there has been significantly increased interest in this biopolymer. PHBV has a high melting point and high crystallization. Due to its high crystallinity, PHBV is stiff and brittle, resulting in very poor mechanical properties. In addition, PHBV is unstable at a temperature close to its melting point, resulting in a significant decrease in molecular weight during its processing [101]. However, various attempts have been made to improve the thermal stability together with the physical and mechanical properties of PHBV, which include blending or filling techniques [102]. The biocompatibility, bioresorbability, and wet electrospinnability have allowed PHBV to be widely accepted as a microbial copolymer that is often used for TE of scaffolds. The advantage of PHBV is that it minimizes the risk of inflammatory reactions in tissues since its biodegradation products are components of human blood. Overall, PHBV has been extensively investigated for skin TE applications as a suitable pro-regenerative material [103]. PHBV is also often combined with ceramic particles for the biological purpose of improving cell growth in vitro (bioactive glass, hydroxyapatite) or antibacterial activity (zinc oxide (ZnO), titanium dioxide (TiO_2_)) [104]. Another advantage of such a hybrid structure is the improvement of the mechanical properties of comparatively weak electrospun PHBV fibers [105]. PHBV is degraded in vivo to hydroxybutyric acid, which is easily metabolized by the body [106].

#### 2.2.7. Hydrogels Based on Natural Polymers

A typical 3D polymer network scaffold called a hydrogel can have an interconnected pore structure that can hold a lot of water and create an environment for cell proliferation, differentiation, and growth that is similar to the ECM [114]. Additionally, hydrogels made from natural polymers, such as polysaccharides (sodium alginate, gelatin, chitosan, hyaluronic acid, and proteins), exhibit good biocompatibility and bioresorbability [115,116]. 

Covalent bonds can be created during chemical synthesis to create bioresorbable hydrogels, or they can be created through physical interactions. Due to these structures’ chemical resilience, the first technique is usually selected. The issue of bulk deterioration and a lack of local flexibility is raised by typically irreversible chemical crosslinking [117]. The interactions between the structural elements of the scaffolding components determine how hydrogels and water-swollen polymers behave. Most biological hydrogel interactions are based on hydrogen bonds, Van Der Waals interactions, and electrostatic interactions. They can be produced by continuously expanding the network system. It is the best method for developing realistic 3D supramolecular hydrogel materials through adaptive design [118]. To create more porous structures, most self-assembled hydrogels can reconfigure their crosslinked networks through continuous development using a combination of hydrogen bonding and intermolecular interactions. The supramolecular design of natural hydrogel materials with features including bioresorbability, biocompatibility, fibrillar architectures, porosity, and hydrophilicity depends on optimizing these interactions [119,120].

### 2.3. Hybrid Biomaterials

The polymer blend (mixture) is obtained by a combination of two or more polymers. Thanks to this technique, new materials with superior physical properties could be prepared. However, the blend’s ultimate characteristics are significantly affected by the miscibility of each component. In addition, miscibility also influences the morphology of separated phases [121]. Polymer blends are divided into three categories:Homogenous polymer blends (thermodynamically miscible polymers): this blend type often consists of polymers with similar chemical composition. Their mixing leads to the preparation of the blends with a single-phase structure and only one glass transition temperature is observed;Compatible polymer blends: this blend type consists of immiscible polymers, but thanks to their strong interphase interaction, the macroscopically consistent physical properties are observed;Heterogenous polymer blends (immiscible polymers): their mixing results in blends with separated phase structures, and two or more glass transition temperatures are observed [121].

Interest in these materials has arisen mainly due to their ability to modify properties and adapt them to the conditions that are required for their application. Another advantage of combining several polymers is the cost. By combining, the costs of the final product can be significantly reduced, and sufficient or even better properties are maintained [122]. The following text focuses mainly on PCL- and PLA-based blends.

#### 2.3.1. PCL-Based Blends 

PCL-based composites are heavily studied and used in biomedical, medical, or pharmaceutical applications more than PCL-based materials alone. Moreover, PCL blends exhibited better mechanical, thermal, or viscoelastic properties [123].

PCL/PLA has been noted as one of the most investigated blends among biodegradable polymer composites, as they have shown improved mechanical and thermal properties of PCL. Various research groups have been particularly interested in fabrication methods [123], overall characterization [124], as well as biodegradation [125]. Nowadays, many studies are oriented toward the electrospinning of PCL/PLA blends [126]. Electrospinning is one of the most commonly used techniques to obtain continuous fibers in the nanometer size range. Electrospun fibers have the potential to be used for drug delivery, wound healing, TE, and regenerative medicine [127]. The porous structure of PCL/PLA scaffolds for TE is favorable and almost necessary. Except for electrospinning, other techniques allow the design of PCL/PLA-based matrices in a porous manner. For example, Wang et al. applied the batch foaming method to create the porous structure of the PCL/PLA scaffold [128]. Another option presents the addition of a component to the blend, which can be extracted after hardening, e.g., by dissolving [129].

Another popular polymer mixture presents PCL with PEG. This polymeric combination appears as copolymers [130,131,132,133,134,135,136,137,138] or as blends [139,140], even though copolymer use seems to be more popular. PCL/PEG copolymers have the potential to be used in drug delivery applications [130,141,142], TE, and regenerative medicine [143,144,145,146], as well as cancer treatment [147,148,149]. The addition of PEG improves the hydrophilicity of PCL, and so cell adherence is also enhanced. Compared to PCL alone, the PCL/PEG/PCL triblock copolymer has been noted to have lower acidity, higher rate of degradation, as well as lower hydrophilicity, thus making it a better alternative to constructing scaffolds for TE applications [147]. PEG may also act as a plasticizer, leading to the improvement of the PCL/PEG-based scaffold morphology [150]. Bioceramic filler can also be added to PCL/PEG copolymers. Liu et al. proved in their study that the addition of PEG and tricalcium phosphate (TCP) in the PCL-based scaffolds significantly improved cell proliferation, adhesion, and osteogenic differentiation [151]. Bioceramic/polymer composite systems have gained importance in treating hard tissue damage by applying tissue-engineered bone grafts [146]. To gain the porous architecture of PCL/PEG scaffolds, supercritical carbon dioxide as a foaming agent has been applied in multiple studies [150,152].

PCL can also be combined with natural polymers. Engineered human skin is commonly fabricated by using collagen scaffolds that often have poor mechanical properties. To improve the strength of collagen-based scaffolds, PCL is blended with collagen and formed into fibers by electrospinning [153]. PCL/collagen blend is a promising material for skin [153,154,155,156,157,158] and vascular TE [159,160,161]. PCL/chitosan mixture could be a promising strategy for designing wound dressings. This polymer combination allows the engineering of antibacterial, bioactive material with good physicomechanical properties and biodegradability [162].

PCL/starch blends present biomaterials with proven biocompatibility and anti-inflammatory characteristics. The dominant agent controlling the final mechanical properties is the composite/component ratio, which makes it possible to design a highly stiff (more starch content) or more flexible and thermoplastic (more PCL content) scaffold. In addition, controllable mechanical behavior, which can adjust different tissues, made this type of blend appropriate for different biomedical applications [163]. Although the PCL/starch blend is biocompatible and biodegradable, attention is currently being drawn mostly to its agricultural application [164], as it presents environmental-friendly material [165,166].

#### 2.3.2. PLA-Based Blends

Despite having better processability, reproducibility, and mechanical properties, on the other hand, crystallization, brittleness, low thermal resistance, and low heat deflection temperature are considered to be major disadvantageous properties of PLA. Hence the formation of PLA-based blends and composites can be considered a solution for overcoming these drawbacks [167]. Several reports are present in the literature documenting the melt blending of PLA with different bio-based or FBPs, such as PCL, PEG, starch, and PHB [168]. One of the most studied combinations for medical applications is PLA/PCL. This blend was mentioned in the text above.

PLA can be blended with various polymers to modify tissue-specific scaffolds. Solid disks, as well as porous 3D structures based on PLA and coated or filled with collagen, were tested for biocompatibility and endotoxin production [91]. The research confirmed the biocompatibility of PLA, as well as endotoxin contamination levels, that were below the FDA limit. In addition, PLA-printed discs supported the growth, spreading, and proliferation of various cell types, such as osteoblasts, osteoblast-like cells, and human umbilical vein endothelial cells.

PLA blending with biocompatible ceramics (via composite fiber or ceramic coating on polymer scaffolds) can result in increased hydrophilicity of 3D printed scaffolds, which provides a better environment for cell adhesion and subsequent migration [169].

Already in 1990, the PLA/PEG block copolymers were characterized as potential materials suitable for drug carriers. K. J. Zhu alleged that the rates of drug release and biodegradation could be tailored by adjusting polymer composition [170]. Currently, this type of material is being studied for its possible application in nanomedicine, e.g., micelles [171]. Nanomedicine is the branch of medicine that seeks to apply nanotechnology, which involves the manipulation and manufacture of materials or devices that are roughly 1 to 100 nanometers (nm; 1 nm = 0.0000001 cm) in size, to the prevention, diagnosis, monitoring, treatment, and overall regeneration of the human body [172]. In addition, PLA/PEG blend could also be potentially used in TE of bone. Moreover, due to the low melting point, PEG in the blend can act as a plasticizer, which affects the processability [173]. It means that this type of material could be adjusted for 3D printing technology, and thus achieve the engineering of patient-tailored bone grafts [173,174,175]. 

PLA/PEG blend was also studied by Scaffaro et al. [176]. In this study, three-layered scaffolds with a pore size gradient were developed by melt mixing of PLA/PEG blend with sodium chloride (NaCl). Pore dimensions were controlled by NaCl granulometry. Finally, the porogen part of the blends (NaCl and PEG) was removed by selective leaching in boiling demineralized water. A similar method was also used in the experiment carried out by Chen et al. [177]. Authors used not only NaCl as a porogen agent but also supercritical carbon dioxide as a foaming agent to fabricate porous scaffold with high porosity. Porogen leaching is the most common method to obtain a porous structure for this type of polymer blend [178,179].

PLA and PHA blends have the potential to be material for commodity and biomedical applications thanks to their unique mechanical and physical properties with balanced biodegradability at the same time (Figure 6). The miscibility of PLA and PHB (the most common type of PHAs) depends on the molecular weight of PLA. While PHB and low molecular weight PLA are miscible, PHB and high molecular weight PLA are not [168,180]. It has also been described that different compatibilizers also influence the mechanical properties of the PLA/PHB blend [181]. Moreover, Ausejo et al. also proved this hybrid biomaterial to be a promising candidate for TE applications. The authors focused on the characterization of the 3D printed PLA/PHB-derived objects, and results showed their favorable mechanical properties, thermal stabilities, and cell viability [182]. However, scaffolds derived from PLA/PHB blends do not have the porous structure necessary for TE. Currently, there are not many studies that focus on this issue. A study conducted by Sartore et al. faced this drawback. The authors added superabsorbent polymer, which was afterward treated with water to achieve the required porosity [183]. Although PLA/PHB blends hold great potential to be applied in the form of scaffolds to repair damaged tissues, more research is needed in this field to adjust the blend’s properties so scaffolds with advanced characteristics can be engineered.

#### 2.3.3. Polymer/Bioceramic Blends

Scaffolds solely composed of polymers often exhibit inadequate mechanical properties, particularly in terms of mechanical strength, Young’s modulus, and toughness. Another challenge is their low bioactivity, which can be addressed by incorporating ceramic materials such as hydroxyapatite (HAp), tricalcium phosphate (TCP), or bioactive glasses. These ceramics interact with physiological fluids and form strong bonds to hard tissues, and in some cases, soft tissues, through cellular activity. The combination of flexible polymers with bioceramics promotes tissue regeneration [184]. Bioceramics belong to a class of inorganic and nonmetallic ceramics utilized for repairing and reconstructing damaged musculoskeletal and periodontal structures. They possess excellent biocompatibility, osteoinductivity, corrosion resistance, and high compressive strength. However, they also have limitations, including a brittle surface, low fracture toughness, reduced mechanical reliability, low elasticity, and exceptionally high stiffness compared to human bone’s cortical properties. Bioceramics exhibit higher strength under compression but are weaker under tension, necessitating careful consideration when designing scaffolds for specific biomedical applications [185]. They are more suitable for cementing substances, reinforcing materials, and implants to repair and replace damaged structures within the skeletal and muscular systems [186]. Ceramic biomaterials often consist of inorganic calcium or phosphate salts that possess osteoconductive (promoting new bone ingrowth) and osteoinductive (promoting osteoblastic differentiation) properties. Hydroxyapatite (HAp, Ca_10_(PO_4_)_6_(OH)_2_) (Figure 7a,b), β-tricalcium phosphate (β-TCP, Ca_3_(PO4)_2_), and bioactive glasses are among the most commonly used biomaterials for 3D scaffolds in bone regeneration [10]. 

The incorporation of bioceramics into polymer matrices produces hybrid/composite materials, which have been extensively studied in the TE of hard tissues such as bone and teeth. A Swedish research team reported that the degradation rate customization relies heavily on the material composition, with the incorporation of mineral phases such as HAp enhancing the degradation of PLA/PCL combinations [187]. Through controlled porous structures and scaffold design, the material can further regulate degradation and facilitate new bone formation according to the patient’s requirements. 

One advantage of combining PLA with bioceramics is the possibility of additive manufacturing, enabling a high degree of personalization [169,188,189,190,191,192]. PLA/bioceramic blends can exhibit superior biocompatibility and osteogenic induction properties compared to pure PLA scaffolds [193]. One extensively studied combination is PLA combined with HAp, which can be processed using low-cost and stable fused deposition modeling (FDM) technology [189]. The addition of PLA to HAp significantly alters the mechanical properties from brittle to ductile fracture, leading to notable improvements in flexural and compressive strength [194].

Nikpour et al. conducted a study on nanostructured bioactive glass/dextran composite scaffolds for bone TE. They demonstrated that the inclusion of bioactive glass nanoparticles within the hydrogel matrix resulted in a significant improvement in mechanical strength. Furthermore, these scaffolds exhibited enhanced reactivity with body fluids, creating active sites for mineralization [72]. To further enhance bioactivity and improve the mechanical properties of the material, inorganic materials such as hydroxyapatite (HAp) were also incorporated into the dextran matrix [195]. Fricain et al. described a composite macroporous material suitable for bone TE, consisting of nano-hydroxyapatite-pullulan/dextran polysaccharide [196].

In general, ceramic-based materials can be combined with biodegradable polymers, including those discussed earlier in this review, for use primarily in bone TE applications.

## 3. Interaction between Biodegradable Scaffolds and Host Immune System

### 3.1. Immunoengineering as an Emerging Field in the TE

Material studies, especially those focused on the scaffold application in the TE, have been persistently developing. Traditionally, the first approach was to construct such a biodegradable polymer-based matrix which should have stayed ‘biologically invisible’ after implantation into the living organism. Nowadays, this strategy does not seem effective as it is known that any in vivo intervention triggers intrinsic pathways leading to the inflammatory response [197]. Based on this knowledge, it is believed that inflammatory-driven tissue regeneration mediated by scaffolds could present a better strategy. In addition, scaffolds should modulate and cooperate with the immune system so that all biochemical processes happening in the affected site lead to tissue repair. For this reason, immunoengineering, as a new scientific field, has emerged [198]. Although many scaffolds have been successfully applied in vivo or utilized in clinical medicine to restore damaged tissues, adverse effects such as extensive inflammatory response leading to transplant rejection were often observed. This can be related to the fact that fabricated materials were mainly developed as biologically inert, so they would not have provoked any reaction from the recipient’s side [199]. However, as mentioned above, successful implantation into the human body or any living organism is closely related to the host immune system’s response towards implanted material. By this mechanism, a transplant could be accepted or rejected [200]. Immunoengineering investigates and modulates each component of TE (materials, cells, and regulatory molecules), considering intervention with the host immune system [198]. It aims to develop easily accessible, resorbable, immunomodulatory scaffolds that would enhance the endogenous pro-regenerative environment in affected sites, in situ [201]. Additionally, immunoengineering also distinguishes that the activity of the immune system, together with the body’s regenerative capacity, greatly differs within individuals. For this reason, sex, age, and comorbidities are also taken into consideration, thus allowing personalized TE [202].

### 3.2. Scaffolds and Foreign Body Reaction (FBR)

When a scaffold is applied in vivo, a series of specific inflammatory and wound-healing reactions are activated as affected tissue elicits cellular response [203]. Depending on the ratio of pro- and anti-inflammatory agents, either extensive fibrosis, in the form of a fibrous capsule, or tissue regeneration evolves from this process [204]. FBR can be described as a stage of chronic inflammation of the tissue, which is related to the presence of foreign material in the body (Figure 8). This phase is characterized by the occurrence of the foreign body giant cells. A detailed description of this phenomenon is described in the following text.

At first, implantation of the scaffold causes iatrogenic injury resulting in the onset of the acute inflammatory reaction. Adjacent capillaries dilate, and cytokines together with damage-associated molecular patterns (DAMPs) are released from platelets, damaged cells, and ECM. Simultaneously, proteins from blood and serum are adsorbed on the scaffold surface, activating the coagulation pathway and leading to the formation of a temporary matrix [205]. This matrix is placed around and on the scaffold, and research confirmed that accumulated fibrinogen is the main initiator of the acute inflammatory phase. This stage is also characterized by the recruitment of neutrophils and by the release of the cytokines IL-4 and IL-13 from the mast cells. The range of their concentration influences the extent and onset of the FBR [205]. Released pro-inflammatory cytokines, chemokines, and oxide radicals recruit other neutrophils, mast cells, and monocytes/macrophages. In addition, secreted signaling proteins also activate adaptive immune cells presented by B cells, CD4+ T cells, CD8+ T cells, natural killer cells, etc. Pattern recognition receptors situated in the innate immune cell membranes recognize the temporary matrix proteins and target the material for phagocytosis [206]. The acute inflammatory phase lasts approximately 3–7 days. Within this time, the immune system tries to degrade implanted material and restore damaged tissue. However, as the scaffold persists in the organism, the onset of the chronic inflammatory reaction begins [203]. This stage is predominantly mediated by macrophages and foreign body giant cells, which are generated from the fused macrophages. In addition, lymphocytes also participate, but their role still remains poorly understood. Macrophages, by crosstalk with the myofibroblasts, regulate the degradation of the implanted material and also coordinate the formation of granulation tissue, which is the precursor for fibrous capsule development [207]. This phenomenon is dependent on the local tissue biochemical environment, which is formed by many cells, including T cells, and determines the range of fibrosis. According to the signal that macrophages receive, they are capable of polarization, or so-called phenotype adaptation [208]. For tissue regeneration, it is crucial that macrophages switch from pro-inflammatory phenotype (M1) to pro-regenerative phenotype (M2). Important is also the switch of T helper 1 (Th1) cells to T helper 2 (Th2) cells. Foreign body giant cells play a crucial role in phenotype adaptation. For the proper interaction between the scaffold and the immune system, pro-regenerative mechanisms must be in the majority [208].

### 3.3. Immunomodulatory Scaffolds

Immunomodulatory scaffolds should have properties that will not negatively interfere with the host immune system and cause extensive scar formation. Conversely, they ought to modulate immune cells together with the degree of inflammation and thus hinder fibrosis of the tissue [209]. That is why antigen, as well as immunomodulatory specificity of the transplanted scaffold, seem to be the key factors for functional and long-lasting tissue repair [201].

Main scaffold properties that need to be adjusted for immunoengineering involve shape, topography, micro-architecture presented by porosity and pore size, stiffness, hydrophobicity, and chemistry of the used polymers [201]. It is also crucial to control degradation kinetics, mechanical stimuli, and oxygen concentrations in the transplanted site [210]. For example, several studies demonstrated that fiber diameter and organization within the scaffold structure influence macrophage polarization and the degree of the inflammatory response [211,212,213]. Results suggested that aligned fibers with bigger diameter sizes were more favorable for M2 polarization. Porosity and pore size represent other scaffold immunomodulatory features. Findings revealed that bigger pore size had a more pro-regenerative effect [214,215]. However, multiple big pores might negatively affect the mechanical stability of the structure, so an adequate balance needs to be found. In addition to crucial scaffold properties, several studies demonstrated that fibrotic capsule was formed around scaffolds with smooth surface [216].

The kinetics of biodegradation also predetermines whether the scaffold triggers an immune response in a pro-inflammatory or pro-regenerative way [217]. Non-degradable materials usually provoke the formation of the fibrotic capsule, which in some cases might be beneficial (artificial joint stabilization). However, this effect is undesirable for the regeneration of soft tissue [208]. In the context of fibrotic capsule formation, scaffolds based on natural polymers seem to have better interaction with the host immune system when compared to fossil-based scaffolds [218]. On the other hand, the degradation of these matrices cannot be precisely controlled, which leads to a discrepancy between scaffold degradation and the formation of the new tissue. Furthermore, the synthetic material’s prolonged presence causes an inflammatory response mediated by phagocytes and foreign body giant cells. By-effects of this phenomenon include fibrous encapsulation of the implanted scaffold [199]. In addition, by-products that are generated in the process of material breakdown have to be studied as well. Released acidic products, in particular, can enhance the inflammatory response in the affected site, supporting tissue fibrosis rather than regeneration. The solution might present the incorporation of basic molecules with the ability to neutralize redundant acidity and maintain pH homeostasis in this biochemical environment [219]. Considering all the aforementioned facts, hybrid bio-based/fossil-based scaffolds might present a good compromise for scaffold immunoengineering.

When predicting immunological response to implanted material, scaffold-independent factors need to be considered as well. These are, for example, anatomical location, age, and comorbidities [199]. The implantation site, as well as the procedure of scaffold implantation itself, affect the recruitment and composition of adjacent cell populations. Studies also pinpointed that younger organisms had higher regenerative capacity, and the inflammatory response, as well as fibrosis, were reduced. Sufficient vascular supply is also required for successful scaffold-driven tissue repair [220].

Material origin (fossil-based or bio-based polymers) also modulate immune response [201]. In addition, the results of a recent study demonstrated that scaffold-driven immune response depended on the chosen polymer-based scaffold [221]. At first, results proved that all implanted scaffolds triggered higher immune responses when compared to the control group (untreated animals). Surface antigen CD206, which is typical for M2 macrophage phenotype, was highly expressed in the group with inserted biological scaffolds. Moreover, expression of the Th-2-associated genes was also detected. These findings indicated that biological scaffolds induced type-2 immune response (i.e., protective immunity). On the other hand, a chronic inflammatory response characterized by large infiltration of neutrophils and depletion of M2 macrophage antigens was observed within the fossil-based scaffold implants. Outcomes also revealed that stiffness and material size determined the scale of neutrophil recruitment. In conclusion, biological scaffolds seemed to have more pro-regenerative potential.

Current modern approaches allow the modification of fossil-based scaffolds in an anti-inflammatory manner, so they can also induce pro-regenerative mechanisms in situ. Cell-seeding method seems to be beneficial in enhancing the biological properties of the synthetic matrices [222]. On the other hand, a graft-versus-host reaction might develop after implantation due to the cellular component [223]. To minimize this phenomenon, the application of cell-free scaffolds loaded with nanosized extracellular vesicles (i.e., exosomes) derived from mesenchymal stem cells has also been a point of interest in various studies [224,225,226]. Zhang et al. developed a 3D PLA-based exosome-loaded scaffold with a porous design in order to achieve immune-driven osteoregeneration [227]. Results obtained from in vitro experiments determined the anti-inflammatory effect of exosome-loaded scaffolds reflected in decreased expression of inflammatory cytokines and production of reactive oxygen species when co-cultured with pro-inflammatory macrophages. Moreover, the addition of these bioactive scaffolds in the mesenchymal stem cell cultures enhanced osteogenic differentiation. Another study investigated composite scaffolds based on PLA and hydroxyapatite [228]. Selective laser sintering was the chosen technique for its fabrication, allowing the engineering of the patient-tailored matrices. In addition, their surface was modified with a hybrid nanomaterial consisting of quaternized chitosan, graphene oxide, and polydopamine. When applied in vivo to repair large bone defects (animal model, rats), accelerated bone healing was observed. Moreover, scaffold modification with nanohybrid components positively influenced the M2 type of macrophage polarization as well as osteogenesis and angiogenesis.

Presented facts considering the application of both fossil-based and bio-based polymers demonstrate that they hold the potential to be applied in vivo for tissue restoration. However, more experiments need to be performed to better understand the divergence of the immune response evoked by various materials and, therefore, facilitate reception by host organisms.

## 4. Conclusions and Final Remarks

The field TE aims to regenerate damaged tissues by utilizing highly porous scaffold biomaterials in combination with cells from the body. These scaffolds act as templates for tissue regeneration, guiding the growth of new tissue. The selection of materials for scaffold manufacturing must adhere to specific criteria, including intrinsic bio-functionality and appropriate chemistry to promote molecular biorecognition by cells, stimulating proliferation, adhesion, and activation. Furthermore, the mechanical properties of the scaffolds and the kinetics of decomposition in the chosen materials must be adjusted according to the specific TE application to ensure essential structural functions and achieve the desired rate of tissue formation. Geometrical features such as exposed surface area, pore distribution, porosity, and architecture influence the rate of cell penetration within the scaffold volume and the formation of the extracellular matrix (ECM). Nanofibers have been more favorable as scaffold biomaterials compared to microfibers due to their nanosize, which promotes cell morphology resembling in vivo conditions. However, fabricating an ideal scaffold that fulfills all desired properties, including biocompatibility, biodegradability, viscoelasticity, mechanical strength, structural stability, antimicrobial activity, clinical simplicity, and cost-effectiveness, remains a challenge.

While the biomaterials discussed in this review offer advantages as scaffold materials in TE, they also have limitations that restrict their individual applications. The combination of multiple biomaterials to create hybrid materials with enhanced functionalities has the potential to overcome the limitations of each material and meet clinical requirements. Therefore, the discovery of new materials and the development of composite scaffolds using existing biomaterials will continue to be a focal point of future research in the field of TE. Despite significant progress in biomaterial research, the development of 3D scaffolds holds promise for improved tissue regeneration, transitioning from animal studies to intended clinical use. Efforts should be directed towards a clear understanding of the interactions between biomaterials and tissues, particularly regarding mechanical strength and fatigue limits under periodic external stress. In the context of regenerative medicine, scaffold-based in situ TE presents a new perspective for the effective restoration of damaged organs [229]. The traditional TE approach utilizes cell-seeded constructs, often loaded with exogenous growth factors or cytokines, to reconstruct the damaged tissue [230]. On the contrary, in situ TE technology relies on the application of advanced, bioactive, cell-free scaffolds, which should trigger the endogenous healing mechanisms right at the affected site. The main goal is to achieve the active self-repair of the impaired organ [231]. This could be gained by choosing the correct polymer for the scaffold fabrication, as the material influences the immune reaction. It is assumed that scaffold-driven immune response further affects the recruitment of endogenous stem cells and regulatory molecules responsible for the healing process [232].

Provided information allows readers to determine which material is best suited for their specific application. However, it should be noted that many of these materials have not been fully optimized for particular TE applications, and further work is needed to optimize their formulations for translation into clinical practice for targeted applications.

## Figures and Tables

**Figure 1 materials-16-04267-f001:**
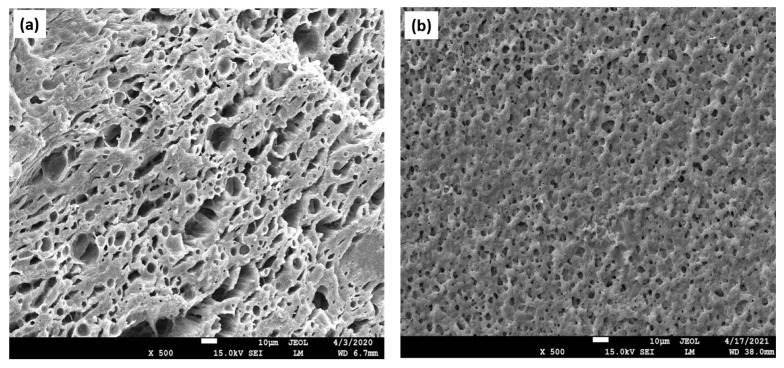
Porous structures of fractured biopolymer-based scaffolds. Scanning electron microscopy (SEM) imaging technique was applied for the analysis of the scaffolds’ cross-sections. (**a**) An image depicting the porous architecture of the matrix. Both larger, as well as smaller pores, are displayed, which are distributed in an inhomogeneous manner; (**b**) observation of the multiple micropores with a homogenous distribution, creating the inner environment for cell attachment, proliferation, and migration, as well as differentiation.

**Figure 2 materials-16-04267-f002:**
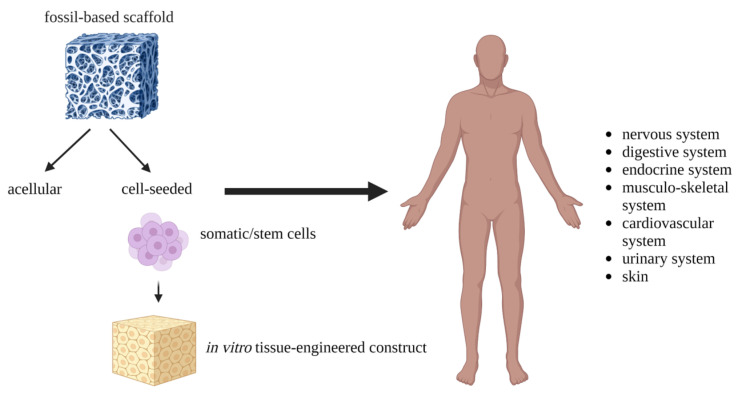
Fossil-based polymers as scaffolding material in TE. FBP might be applied in the form of acellular or cell-seeded constructs to restore damaged organs. However, because of the great regenerative potential, current studies are mostly focused on effective scaffold colonization with stem cells. Created with BioRender.com.

**Figure 3 materials-16-04267-f003:**
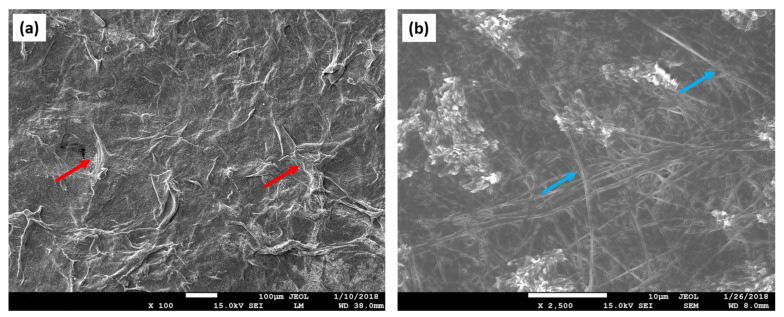
SEM images revealing the surface topography of the collagen scaffold. (**a**) Collagen fibers (red arrows) create the rough surface of the scaffold. Depicted design enlarges the overall surface of the matrix, thus providing a larger area for the cell attachment; (**b**) a detailed analysis of the scaffold’s architecture displays collagen fibers oriented in an unaligned manner (blue arrows).

**Figure 4 materials-16-04267-f004:**
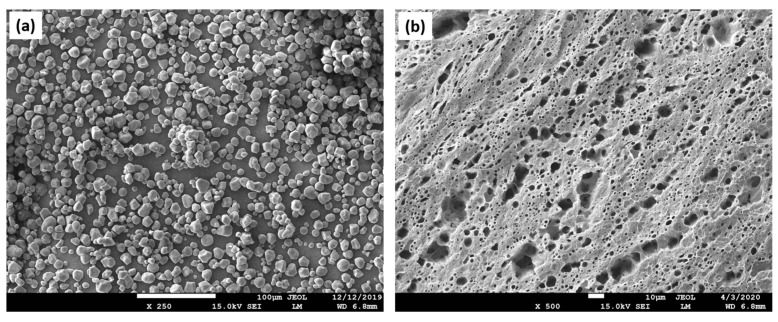
Starch as a natural polymer for scaffold engineering. (**a**) SEM image depicting dispersed starch granules. These units can modulate both topography (surface roughness) and the inner design of the scaffold; (**b**) when used as a porogen added to a polymeric blend, starch granules can create a porous structure.

**Figure 5 materials-16-04267-f005:**
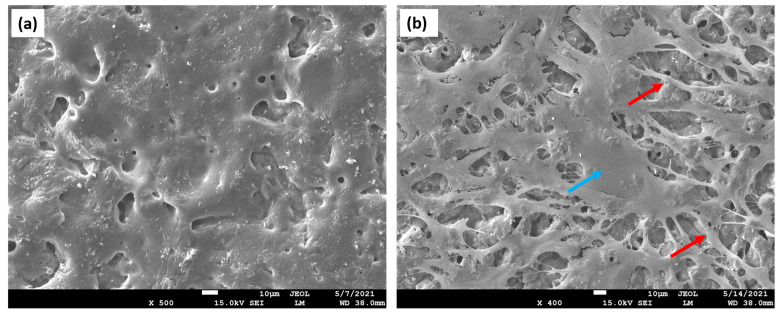
PLA-based scaffold. (**a**) Surface analysis of the PLA scaffold engineered by the pressing technique (reference sample). SEM image displaying smooth, homogenous area for the cell attachment; (**b**) PLA-based scaffold seeded with human fibroblasts (blue arrow). After 7 days of incubation, a dense cellular layer on the scaffold surface could be observed. Multiple filopodia (red arrows) are also detected, proving efficient attachment to the scaffold surface. The presented findings demonstrate good scaffold-cell interactions and determine good biocompatibility of the material.

**Figure 6 materials-16-04267-f006:**
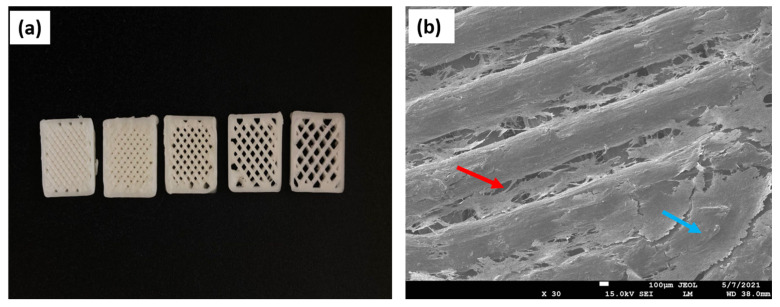
PLA/PHB blend in the form of 3D scaffolds. (**a**) Three-dimensional printed scaffolds with different designs; (**b**) SEM image displaying adipose tissue-derived mesenchymal stem cells used for scaffold colonization (blue arrow). Cytoplasmic protrusions and filopodia (red arrow) enabled effective cell attachment to the scaffold surface. Observed coherent cell layers indicate adequate scaffold-cell interactions.

**Figure 7 materials-16-04267-f007:**
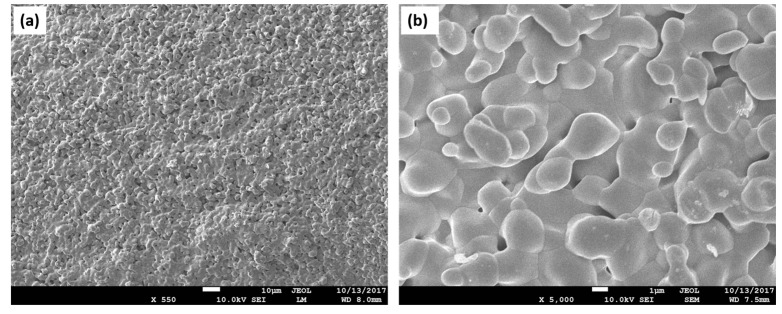
SEM images of the HAp-based scaffold. (**a**) Topography analysis depicts the rough surface of the matrix; (**b**) the inner structure of the fractured scaffold. A dense matrix mass can be observed, which could be beneficial for its application as a cell-free scaffold for TE of the hard tissues. In addition, HAp is also often combined with other polymers to create hybrid materials with enhanced properties.

**Figure 8 materials-16-04267-f008:**
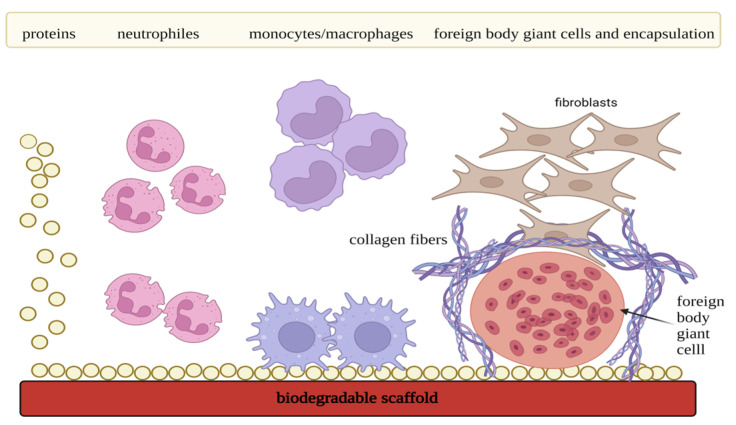
Foreign body reaction. Simplified scheme of the foreign body response development. After the scaffold implantation, the material interferes with the blood/serum proteins, which are adsorbed on the surface. Subsequently, neutrophils, the cells of the innate immune system, are activated. By secretion of the regulatory molecules, macrophages are recruited, and with the material’s persistence in the body, foreign body giant cells are formed. These cells are typical for foreign body reactions. Macrophages crosstalk with (myo)fibroblasts, forming a fibrous capsule around the implanted material. Created with BioRender.com.

**Table 1 materials-16-04267-t001:** Summarization of the most common characteristics of fossil-based polymers.

Polymer	Advantages	Disadvantages	TE	References
Poly(ε-caprolactone)	bioresorbable biocompatible mechanical stiffness elasticity thermal stability low inflammability	low cell adhesion slow degradation	bone dental retina skin, vascular liver cartilage ligaments muscle, neural	[12,16,17,18]
Poly(vinyl) alcohol	biocompatibility hydrophilicity permeability water solubility	biodegradability after crosslinking cell adhesion	cardiac vascular bone skin cartilage neural corneal	[14,16,19,20,21,22,23,24,25]
Polyethylene glycol	biodegradability non-ionic water-soluble thermally stable protein repellency	cellular adhesion antigenicity	bone drug delivery	[16,26,27,28,29,30,31,32]
Polypropylene fumarate	biodegradability, biocompatibility excellent strength osteoconductivity	hydrophobicity	orthopedic, neural bone, ophthalmology cardiac	[33,34,35,36]
Polyurethane	biodegradability biocompatibility flexural endurance thrombo-resistance oxygen permeability	low interfacial tension	cardiac bone cartilage	[16,37,38,39,40,41]

**Table 2 materials-16-04267-t002:** Basic characteristics of natural polymers.

Polymer	Advantages	Disadvantages	TE	References
Chitosan	biocompatibility bioresorbability physiologically degraded	mechanical properties resistance to enzymatic degradation	skin bone	[44,45,46,47,48,49]
Collagen	biocompatibility bioresorbability cell interaction	mechanical properties rapid degradation immunogenicity	skin bone	[50,51,52,53,54,55,56]
Hyaluronic acid	biocompatibility			
	bioresorbability		bone	
	enhances cell proliferation	poor cell adhesion	muscle	[57,58,59,60,61,62,63]
	immunosuppressive antioxidative properties			
Fibrin	biocompatibility	mechanical properties	skin	
	bioresorbability cell interactions common natural protein	scaffold contraction	cardiovascular musculoskeletal nerve	[64,65]
Xanthan gum	biocompatibility	mechanical properties	soft tissues	[66,67,68,69]
	bioresorbability	difficult processing	bone	
Dextran	biocompatibility	high cost	skin	
	bioresorbability antithrombotic properties easily derived	low availability	vascular	[70,71,72]
Starch	biocompatibility bioresorbability low cost	dimensional stability mechanical properties difficult processing high water uptake	bone	[73,74,75,76,77,78]
Poly (lactic acid)	biocompatibility		cardiovascular	
	bioresorbability mechanical strength processability	acidic degradation by-products degradation rate low cell adhesion	bone skin tendon	[79,80,81,82,83,84,85,86,87,88,89,90,91]
Polyhydroxybutyrate	biocompatibility	crystallinity	skin	[92]
	bioresorbability natural human metabolite	brittleness hydrophobic thermal stability	bone cardiovascular	[93,94,95,96,97,98,99,100]
Poly(3-hydroxybutyrate-co-3-hyxdroxyvalerate)	biocompatibility	fragility		
	bioresorbability mechanical properties flexibility	impact resistance hydrophobicity thermal stability	bone cartilage	[101,102,103,104,105,106]

## Data Availability

Data sharing not applicable.

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
