# Peer review of "Resorbable Biomaterials Used for 3D Scaffolds in Tissue Engineering: A Review"

_materials, 2023, doi:10.3390/ma16124267_

Round 1
Reviewer 1 Report
Minor revision
1) Title: ok
2) Abstract: is more general; please specify exactly the area of the review position, if you can add some numbers (100% compatible or less?).
3) Keywords: add biopolymers instead of polymers (if you are using bio-product if synthetics doesn’t matter)
4) Introduction: poor references, please add some others to shows exactly the art state of this topic
5) About fossil polymers: any information about their toxicity?
if so please remove them (the first condition they should to be non-toxic).
6) Conclusions and Final Remarks (add your perspectives)
7) References an update by adding 2023
With regards
Author Response
Dear Reviewer 1,
On behalf of all authors of the presented manuscript titled "Resorbable biomaterials used for 3D scaffolds in tissue engineering: A review", we would like to thank You for the time and effort that You have dedicated to provide your valuable feedback on our manuscript. We appreciate your insightful comments which helped us to improve the quality of our paper. We have been able to incorporate changes to reflect most of the suggestions provided by You. Please have a look at the attached document which includes point-by-point response to your comments and concerns.
We look forward to hearing from You and to responding to any further questions and comments You may have.
Yours faithfully,
Martina Culenova

Reviewer 2 Report
line 69.- Change Focued for Focused
Figure 1.- Described the differences between figure 1a and 1b
line 81.- Change natural gass for gas.
line 94.- Change utilised for utilized.
Figure 2.- change focused and colonization.
line 159.- separete inmunogenicity and "and".
line 175.- Aslo for also.
And... A lot more; please check them.
Because your work is a review, you should consider and add the molecule of every polymer you explain.
Figure 3. Describe better the differences between both figures.
You should considerate the separation of chitosan and fibrin; by the way, is chitin not used as a cellular scaffold?
Overall, please describe better the differences between all your figures.
There are a lot of things that could be improved in the writing, I made just a few changes, but it takes a lot of time to waste checking the grammar errors instead of the scientifical soundness.
Author Response
Dear Reviewer 2,
On behalf of all authors of the presented manuscript titled "Resorbable biomaterials used for 3D scaffolds in tissue engineering: A review", we would like to thank You for the time and effort that You have dedicated to provide your valuable feedback on our manuscript. We appreciate your insightful comments which helped us to improve the quality of our paper. We have been able to incorporate changes to reflect most of the suggestions provided by You. Please have a look at the attached document which includes point-by-point response to your comments and concerns.
We look forward to hearing from You and to responding to any further questions and comments You may have.
Yours faithfully,
Martina Culenova

Reviewer 3 Report
The review article by Sara Agocsova and others, entitled “Resorbable biomaterials used for 3D scaffolds in tissue engineering: A review” summarize key biomaterials used as scaffolds for tissue engineering applications. The following comments need to be address by the authors:
1- In page 2, line 47: The authors stated: “Scaffold is a three-dimensional (3D) mesh”. This is a very week and narrow definition for “scaffold”. Please consult the literature and give a more accurate definition.
2- The English quality needs to be improved and use more scientific terms.
3- The authors must carefully edit the text and pay attention to the many typo and formatting issues. Few examples: In page 1 line 22-23, in the abstract section. There is a formatting issue. In page 5, line 159: There is a typo, space is missing between in “immunogenicityand”. Also, line 165 an extra full stop after “units. .”.
The English quality needs to be improved and use more scientific terms.
Author Response
Dear Reviewer 3,
On behalf of all authors of the presented manuscript titled "Resorbable biomaterials used for 3D scaffolds in tissue engineering: A review", we would like to thank You for the time and effort that You have dedicated to provide your valuable feedback on our manuscript. We appreciate your insightful comments which helped us to improve the quality of our paper. We have been able to incorporate changes to reflect most of the suggestions provided by You. Please have a look at the attached document which includes point-by-point response to your comments and concerns.
We look forward to hearing from You and to responding to any further questions and comments You may have.
Yours faithfully,
Martina Culenova
